# Analysis of Religiosity in Slovakia since 1989 and Paths of Its Development

Peter Kondrla [1,*], Andrea Leskova [2] and Eva Durkova [1]

1   Institute for Research of Constantine and Methodius' Cultural Heritage, Faculty of Arts, Constantine the Philosopher University in Nitra, 949 01 Nitra, Slovakia
2   Department of Esthetic and Ethics, Faculty of Arts, Constantine the Philosopher University in Nitra, 949 01 Nitra, Slovakia
*   Correspondence: pkondrla@ukf.sk

**Abstract:** The study presents the results of research into the current state of selected religiosity phenomena in the Slovak Republic and models possible pathways for the future development of the selected phenomena. The starting point is an analysis of three decades of development of religiosity in the region from 1989 to the present day. The input data were obtained as part of a research project focused on development prospects. The data refer to the second and third decades of the reporting period. The data are subjected to statistical and correlation analyses. Analyses deal with the phenomenon of faith and its content, rational and liturgical interpretations of religiosity, and questions of the meaning of existence and the moral dimension of faith. Thanks to the application of selected religionist theories, this formulates possible paths which the religiosity of the selected region will take, describing the potential as well as the risks of development for individual directions of development.

**Keywords:** religiosity; Slovakia; development; changes; perspectives

## 1. Introduction

For a variety of reasons, the Slovak Republic constitutes a specific religious environment. The first reason is its totalitarian government experience, followed by reference to Cyril and Methodius' tradition as a cultural and religious foundation (Hetényi and Ivanič 2021). During his papacy, Pope John Paul II, a Pole, gave Slovakia a lot of attention. There is a well-developed Marian cult, also strongly supported by John Paul II. Folk piety, which comes in a variety of forms, plays a significant role in religion. Folk piety emerged in the period between the two world wars and has since become an essential component of people's cultural lives. This mostly involves pilgrimages or celebrations that include various sorts of entertainment. In light of Central Europe's post-war transformation, these basic forms of popular piety have remained set in the minds of a considerable portion of the population as religion (Babbie 2004; Beyer 2005). The countries of Central Europe were ruled by communist governments controlled from the Soviet Union's headquarters from the end of WWII until the beginning of the last decade of the twentieth century. The communist parties' ideological framework was materialistic, and one of its goals was the abolition of religious faith and religion as a whole. Materialism was founded on a reading of Marx's intellectual legacy, according to which religion was nothing more than mankind's opiate, a form of pain relief. Social injustice, according to Marxist ideology, is the foundation of capitalism and the source of man's suffering. Removing religion would mean that individuals no longer require treatment for misery because they are no longer in pain and do not live in an unequal society. While communist-dominated governmental systems have not eradicated social injustice, they have tried to abolish religion to satisfy their ideological ambitions. Ideologists forbade individuals from attending church services and persecuted church representatives and residents who publicly displayed their faith and affiliation

with a church. The shift happened in the last decade of the previous century, when the political systems of Central European countries became more stable (Tirpák 2017). Totalitarian regimes were replaced by democratic governments, and religion adapted to fit new circumstances. As part of our research, we will track the development of religiosity from this point forward. Religionists and sociologists of religion who specialize in the Central European region discuss three major patterns in the evolution of religiosity in this region (Piwowarski 2000). Štefaňak observes that stereotyped notions of religious metamorphoses in Slovakia are not far from the truth. His study examined respondents' perspectives on how they rate their religion today and 10 years ago. "It is true that, when comparing the current situation to the situation roughly ten years ago, the religiosity of the examined Slovaks has decreased slightly—this is proven mostly by their own pronouncements in the appraisal of their own religious modifications. Remember that respondents were nearly twice as likely to declare its weakening (26.6%) as its deepening (15.4%)—even though the most common was unaltered (44.6 %). Based on the empirical indicators discussed above, it can be stated that many have confirmed the model of religious stabilization in some models of progressive secularization and some models of religious revitalization" (Štefaňak 2021, p. 76). The results of that research confirm not only the model of religious stabilization and the model of progressive secularization (e.g., a decrease in indicators of the frequency of participation in worship or sacrament of reconciliation, so-called church religiosity or involvement in religious associations and movements), but also, partly, a model of religious revitalization (e.g., a slight increase in indicators of faith in the existence of God, the meaning of prayer, the monitoring of religious televisions or websites, and faith or belief in God's help in difficulty), a model of religious or world-view pluralization (e.g., a slight increase in indicators of faith in reincarnation or the search for sensory values outside of faith), and a model of religious syncretism (e.g., an increase in the religiosity indicator "in its own way"—selective, private, or syncretic) (Štefaňak 2021). The goal of our inquiry is to see if the aforementioned processes of religious revitalization can be recognized in Slovak society and if any sort of syncretism or other means of realizing one's own faith is part of this revitalization. We have been tracking these trends since 1989, when there were democratic reforms in society and the reintroduction of religion to daily life.

## 2. Results

### 2.1. Post-Totalitarian Decade

This is a decade that lasted from 1990 to 2000. According to our analysis, this was an era of profound transformation. Instability was created by citizens becoming acquainted with democracy and confronting numerous unpleasant sides of freedom, such as political power abuse, political collusion with the mafia, intimidation, and so on. The increase in crime, violence, and other bad phenomena that has accompanied the contact with freedom after 40 years of tyranny has challenged trust in mankind and human freedom. In terms of religion, we could describe this time as an endeavor to restore the pre-war partnership of the state and the church. This was a situation in which the church wielded political power, had a large effect on education and culture, and interfered ideologically with societal government. This trend was strongest in Slovakia when compared with other Central European countries, because Slovakia had a clergy–fascist state during WWII, the so-called First Slovak Republic. That is, a state in which the president Jozef Tiso was a priest, and this politician was subservient to fascist Germany. After 1989, Christian political parties were formed to try to create the impression that Christianity was returning to society as a morally restorative element that brings moral principles and order to society. Many wanted the return of Christianity, which would have had political power, similar to what was the case in the so-called First Slovak Republic. These were mainly parties with a nationalist orientation, such as the Slovak National Party, but also other political parties. In addition to the national revival, these parties tried to revive the idea of pan-Slavism, which was a continuation of the Russian ideology of the communist regime. This idea still appears in the political and conspiracy dictionary today. In the first decade of the period of freedom, also

thanks to the experience of the pre-war and war periods, priests were perceived as social authorities. Their authority relied on the historically clear notion that a priest is an educated man who is guided by moral values (Akimjaková and Tisovičová 2017). That is why he is respected by people, even non-believers. The rejection of contemporary and modernizing ideas, which were still referred to as Western ideology, was a defining feature of this period. Western civilization was connected with life based on the concept of consumption, of disordered capitalism, where profit and exploitation of man were the only determining factors. This misinterpretation by Slovak citizens created the perception that Slovakia does not require international cooperation or foreign ideas or capital. We do not want foreigners, we save our own, was their credo.

They were known as the keepers of tradition and traditional ideals. Many of these "keepers of ancient and traditional values", it was later revealed, aimed to retain Russia and its ideology's influence in Central Europe. They are still present in Slovak society, especially on the political scene. They have utilized religion as a symbol of traditional values for this political goal (Ivanič 2019). The ideology of the West, according to the defenders of "traditional values", was associated with liberal theology and hence opposed to official churches in Central Europe. The fundamental reason for abandoning liberalism was the absence of dialogue. The church's communication with the faithful, both openly and internally, did not adhere to the principle of dialogue. Christians, like politicians at the time, despised difference and opposing viewpoints. Christianity has gained prominence in the media and public schools as a result of its collaboration with the new political leadership. The Catholic Church and various Christian denominations traded positions. Religious congregations that were rejected and persecuted by the communist regime have risen to prominence as major political forces. Churches and political figures seeking regime restoration have found themselves in the same boat. There is only one truth, one nation, and one morality. Christianity helped to create a community with a single objective and a single leader. As previously said, this was a return to World War II and the clerical system as circumstances permitted. Religion and politics helped each other to maintain their social status. It was extremely difficult to promote new religious movements and churches in an atmosphere of one faith and one nation. Because they were and still are financially subsidized by the state, official churches actively affected cultural policy. The process of registering a new church, which began during this time period, is so strict that Islam is still not recognized as a religion by the state in Slovakia. New religious trends were restrained in Central Europe; the mocked activities of new religious movements were curtailed or even considered socially hazardous. Based on the foregoing, we could characterize the first decade of freedom as a period of religious return to tradition. This is a distinct attitude that seeks the future of religion by preserving the old and original doctrine, which cannot be altered. Some scholars refer to this tendency in religiosity as a paradigm of "religious fundamentalism", which manifests itself in the concept of "returning to roots"—whether in the pursuit of religious orthodoxy, faithfulness to old religious activities, or traditional moral ideals (Štefaňak 2021). Pilgrimages, festivals, and public worship have all grown in importance as religious rituals. National and religious congregations melded, politicians attended religious celebrations, and church leaders appeared in public with politicians. In the background, there were battles over the church's assets, authority control, and the private businesses of church leaders. At the start of the decade, religion had the quality of spontaneous delight with earned freedom. By the end of the decade, it had already become formalized in the shape of grandiose liturgical rituals that emphasized power and domination (Maturkanič et al. 2022). An increase in believers was a distinct feature of this time period. Following the establishment of democracy, believers who wanted to remain anonymous, avoid communist persecution, and refrain from approaching the sacraments and participating in liturgical events joined the official churches. This increase in the number of believers was also a symptom of the church's growing strength, which eventually began to be codified as a result. Large celebrations and mass events emphasized the individual's official affiliation with the majority rather than his spiritual progress

(Murgaš et al. 2022). The result of formalization was a situation in which the believer stated his allegiance to the church and the church provided him with services for this, such as baptism, marriage in the church, burial in a Christian cemetery, and so on. Only Christians who stated their faith openly were considered believers. Other religions were mocked and discriminated against. This began to change with the advent of the second decade, which was connected with the recognition and eradication of faults. Rulemaking has also begun, as have changes in religious attitudes.

### 2.2. Faith Shifts in the Second Decade

The second decade—the first ten years of the new millennium—was included in the sociological research we did in 2020. As a result, we have enough data to develop new theories or confirm existing ones. Respondents recalled how they lived their faith during this time period and how they viewed their religiosity in retrospect. We did not conduct the research survey directly from 2000 to 2010, but rather in 2020. As previously stated, respondents were asked to compare how they perceive their religiosity today and ten years ago. Before delving into respondents' attitudes, consider the fundamental facts that have influenced this decade and the religiosity within it. The second decade can be described as an era of opening up. During this time, Central European countries joined the European Union and, later, the Schengen area. This also meant true openness, which meant the elimination of borders and border crossing points for European countries. The overall political situation has been influenced by an awareness of the threat of terrorism, to which the Central European countries' weak economic situation has added. Low economic fitness was linked to closure in the previous decade, isolation from international structures, and a tendency toward nationalism. Rejection of all foreigners, including their religious expressions and religiosity, has resulted from economic and political closure. Nationalism became associated with economic backwardness and aggressive attitudes toward otherness of any kind in the second decade. It has a negative connotation in society, but it still wields significant political power today (Kobylarek et al. 2022). Because of integration processes, many young people have gone to work abroad. They gained experience with new ways of thinking, different value orientations, and new forms of religiosity. They were able to compare and assess the impact of the autocratic regime's isolation on their lives and applications. They criticized the traditional values associated with their conservative stance, as well as their inability to respond to changing circumstances. Conservatism has come to be associated with a non-transparent society in which authority decides and imposes its ideas on other subordinate members of society. This opaque system also included clientelism and corruption, which were prevalent throughout society (Akimjak and Račková 2018). Part of the experience with an open society has played an important role in Central European political and social change. Equality of opportunity, respect for others, and open communication resonate with people of all ages. Openness has, logically, begun to manifest itself in the realm of religion as well. According to religionists, this is a period of pluralism of religiosity, as well as a period of loss of faith or abandonment of traditional forms of religiosity (Leskova and Mahrik 2019). This period followed a previous revival of traditional faith and religiosity. "Globalization" has resulted in the gradual internationalization of political structures and economic recovery. Ideas and capital from other countries are no longer viewed negatively. Part of public opinion has come to believe that the truth does not have to be limited to one thing, and that the owner of the truth does not necessarily have power. Churches had to change their attitudes and redefine their role in society as a result of these changes. Churches needed to clear their names so that they were no longer associated with political power, the mafia, or corruption, and could be respected as moral representatives once more. As previously stated, two religious approaches predominated in the second decade of the period under consideration. The first was a conservative approach, while the second was based on pluralism. The conservative approach was manifested in an effort to preserve the church's formal place in society, and the church was to govern society and enter into governance according to this approach.

This strategy aimed to maintain the distribution of political and power forces from the first decade. According to the results of our survey, people who are nationalistically oriented, less educated, or live in the countryside favor this approach. Political incompetence was less visible in the countryside, and there was a kind of idealistic symbiosis. The departure of young people to work abroad or in larger cities has undermined and continues to undermine rural idealism (Petrovič and Maturkanič 2022). The number of supporters of religiously conservative interpretations has decreased. The lifestyle shift was not in favor of traditional values. Conservative believers had no clout in society, and the church had no say in politics. That is why many conservative believers have converted to formal Christianity. In practice, this means that believers live according to religious norms on the outside but do not share them internally. Worshippers attend worship services every Sunday, go to confession, and participate in church festivals, among other things. On the other hand, the same believers defraud businesses, cut taxes, lie, and so on (Petrovič and Murgaš 2020). A situation developed in which churches were grateful for such believers, and, in the second decade, a very strong group of believers with a dual morality emerged. The man of faith was not the same as the man who lived in everyday reality. On the one hand, they discussed the traditional family and the traditional values that people should return to. On the other hand, they did not reflect any of these traditional values in practice and perceived those who did as archaic and outdated.

The second trend, in addition to attempting to preserve conservatism, was to promote religious pluralism. Openness and pluralism of opinions resulted in a critical attitude toward churches, among other things. Criticism of formalism and the moral ambiguity of church practice began to emerge. Churches ceased to be moral guarantors. The criticism focused on church leaders' moral misdeeds as well as their collaboration with authoritarian leaders. This is an ossified attitude toward formal conservatism in the context of the second decade of the period we are studying. Pluralism-based religiosity rejected formalism as a lie, as an attempt to return to the past, dictatorship, and authoritarianism with not only spiritual but also political power. That is why some have argued that churches should not be tied to property or political power, but rather serve as a model of Christian living. The basis of faith is not a religious declaration of faith, but rather acts that express faith (Judak et al. 2022). While allowing for the pluralism of religious opinions and attitudes did not result in protests or insurgencies within churches, it did create pressure to which churches responded. This pressure was the decline in church attendance and the number of worshippers declaring their church affiliation. The number of people interested in church services, priesthood, or a consecrated life increased as well. The Catholic Church was particularly affected by this trend. In the second decade of our reporting period, church leaders began to recognize the importance of self-reflection and opening up to social challenges. They perceived that conservative and radical believers remained in the church. According to our findings, the majority of active believers and the younger generation sought to meet their religious needs outside of the church. This period is distinguished by a slow starting of opening. The church's openness to pluralism was cautious at the level of declaring changes. However, declaring openness was significant, because it was fully developed in the following decade by the emergence of new movements, particularly youth activities. All activities in the second decade had to be carried out under the supervision of churches, or they were considered heretical or dangerous (Rychnová et al. 2022). In general, the second decade was marked by a declaration of openness to the needs of believers. There was a stated willingness to give the faithful space for their own activities, but they were still subject to church supervision. This paved the way for the genuine incorporation of pluralism into the lives of believers, as well as the creation of space for practical secularization.

**Theorem 1.** *The decline of believers in official churches signaled a search for alternative ways to meet spiritual needs, rather than a loss of faith.*

There is a stated willingness to give the faithful space for their own activities, but they are still subject to church supervision. This has paved the way for the genuine incorporation of pluralism into the lives of believers, as well as the creation of space for practical secularization.

In Table 1 we can see in two major trends based on the data presented above. The first is an increase in the number of believers between 1991 and the end of 2001. It is the first decade in which the number of declarations has increased due to a change in the political system. We labeled this period as one of the return of traditional faith. The following decade, 2001–2011, saw a partial decrease by 8%. In 2011, there were more people expressing faith than in 1991, but fewer than in 2001. We saw a 6.2-percent decrease between 2011 and 2021.

**Table 1.** Total number of believers and non-believers.

| Year | Believers | Non-Believers |
|------|-----------|---------------|
| 1991 | 72.8 | 9.8 |
| 2001 | 84.0 | 12.9 |
| 2011 | 75.9 | 13.4 |
| 2021 | 69.7 | 23.7 |

**Proof of Theorem 1.** We partially confirmed Theorem 1. We found that the number of people declaring their faith increased between 1991 and 2001. From 2001 to the present, it has been declining. We relied on Štefaňak's assumption, which describes this trend as a slight decrease with persistent interest. The greatest increase in the number of believers occurred in the first decade, while the greatest increase in non-believers occurred in the last decade. □

The first theorem is not fully established. Based on our research, we cannot confirm whether the 23.7 percent of respondents who declare themselves to be non-believers practice no religion. Incomplete confirmation of the theorem may also result from respondents identifying a faith affiliated with one of the churches or religious movements. They are non-believers simply because they do not belong to any community and describe themselves as such. Their attitude may be influenced by their unwillingness to identify with any religious community. However, we lack sufficient evidence to back up this claim.

**Theorem 2.** *Changes in religious attitudes occurred as a result of churches' openness.*

The formulation of this theorem is based on Štefaňak's previous one, which was published in the work *Religiosity of Youth in the Process of Transformations* (Štefaňak 2019). In his interpretation of his research, he argues that pluralization, which is associated with a revival of interest in religion and faith in Christian churches, is one of the likely scenarios for the possible development of religiosity in Slovakia. According to the data we obtained, there has been an unquestionable increase in the number of non-believers in society over the last decade, as confirmed by the Slovak Republic's Statistical Office. According to the findings of our study, less than half of the respondents did not change their attitude toward faith. Those who have changed their attitudes negatively outnumber those who have changed their attitudes positively.

At the same time, it is important to note that 44.6% of respondents say their attitudes toward faith have not changed. This means that the above volume of respondents' attitudes has not changed in the last decade, and the figures in Table 2 refer to a group of 55.4% of responders who have changed their attitudes toward faith. Furthermore, it should be noted that 15.4% changed their attitude to faith positively and 26.6% changed their attitude to faith negatively. Church experience, or information learned about the church from the media or other sources, was the most significant factor that caused negative changes in attitude toward faith. Positive changes in attitudes toward faith, on the other hand,

occurred as a result of personal experience, which includes unexpected healing or other unexpected situations that respondents interpreted as transcendental interventions.

**Table 2.** Reasons for changing attitudes towards faith.

| Change in Attitude | Church Scandals | Information Experience | Live Events | Personal Experience | Dogmatic Education |
|---|---|---|---|---|---|
| Positively | 0.0% | 1.9% | 20.0% | 26.0% | 0.0% |
| Negatively | 25.5% | 19.8% | 14.8% | 12.0% | 7.1% |

**Proof of Theorem 2.** The first and second parts of the theorem can be partially confirmed. Based on the data in Table 1, we can conclude that the population's attitudes toward faith have shifted. We have demonstrated that there has been a gradual but significant increase during the period under consideration as well as a gradual decrease in believers since the second decade (Tkáčová et al. 2021a). The increase in numbers has been attributed to changes in political conditions as well as the elimination of persecution and intimidation of people who have declared their faith. As long as we follow the believers as a whole, the evolution of the number of believers in churches mirrors the evolution of the number of believers in general. In the first decade, the number of believers grows, but, in the following two decades, the number of believers declines. In the last decade, the total number of believers fell by 6.3%, while the number of Catholic believers fell by 6.2%. The reversal cannot lead us to believe that the number of Catholic Church believers has now stabilized. When compared with other Christian churches, the Catholic Church is losing followers. Some churches have slightly increased their membership, but Buddhism, Islam, and Hinduism have also increased their membership. These are small numbers of believers when compared with the Catholic Church, but we can conclude that a revitalization and revival of interest in faith has been present in the Slovak population over the last decade, but this does not concern the largest churches, which are Catholic and Evangelical A.C. churches, which have been losing believers since 2001. As shown in Table 2, the most common reasons are church dissatisfaction, self-discovery, experience, and life events. These data suggest that faith is being revitalized in society, but this does not apply to the Catholic or Evangelical A.C. churches (Maturkanič 2021). □

### 2.3. Stabilization in the Third Decade

The second and third decades cannot be precisely defined symbolically, but the gradual loss of support for the Christian Democratic Movement in Slovakia attests to changes. The movement narrowly made it into parliament in 2012, but its influence waned after that, and they no longer received enough votes in the next election to be considered an extra-parliamentary party. Religion and spiritual life were no longer associated with Christianity and the traditional understanding of religiosity as something associated with the old liturgy and hierarchical division of functions and tasks. Churches withdrew gradually from public space, particularly in the Czech Republic, but less so in Slovakia. In Poland, the Catholic Church has retained the most clout in public life and remains most visible in the context of its relationship with state power (Tkáčová et al. 2021b). However, it should be noted that laicization has been on the rise over the last decade. Christianity has lost its monopoly on deciding what religion is and is not, what is acceptable from a religious standpoint and what must be rejected in Central Europe. Being an atheist is no longer as strange as being a Buddhist or practicing Indian meditation (Ambrozy et al. 2020). In religious terminology, the period from 2010 to 2020 could be described as a period of revitalization or revitalization of faith. Of course, this is not a renaissance in terms of the return of traditional religious forms. People will not return to churches for traditional worship if interest in faith is restored. Traditional forms of religiosity, as relics, have persisted in the form of pilgrimages or folk piety. Instead of being a response to the needs of young Christians, these activities were primarily aimed at the development of political capital (Judák 2021). Reviving interest

in faith and religion has focused on individual rather than social needs. Religion has moved significantly into public space in the previous two decades, with religiosity occurring as part of social cooperation and communication, but churches have closed themselves off and refused to communicate with the surrounding area. Religion, on the other hand, has moved in the last decade to a private space in which one freely chooses his religious orientation, as well as the subject and form of his religious self-realization.

Religion has emerged as a new phenomenon to which the advancement of communication technologies from various perspectives has also contributed (Kobylarek 2019). The main reason was information flow. Several respondents stated that they had changed their attitudes toward the church as a result of scandals that had come to light as a result of the media. New forms of faith, a resurgence of interest in paganism, and the revival of old cults would not have been possible without adequate information in the media and on social media (Hetényi 2019). The data are only one side of the equation. Social networks, on the other hand, have made communication possible and easier, particularly for marginal religious groups with few members. These members can be located all over the world and still be present in online space at the same time without leaving their homes. The resurgence of interest in religion over the last decade can be attributed to the previous liberalization of social space. The search for spiritual guidance, experimentation with alternative religious practices, meditation, and even religious rap became acceptable, and the concept of religion became very rich in content. In contrast to the first and part of the second decade observed, when the term "believer" meant a man was a Christian (Casanova 2005), the term "believer" referred to a wide range of religious attitudes and opinions in the third decade, including philosophical, nihilistic, or pessimistic attitudes (Akimjak et al. 2022; Pavlíkova and Mahrik 2020). As an example, consider the growing interest in esotericism. There was a surge of interest in Nordic mythology at the end of the last decade. There appears to have been an interest in Slavic pagan rituals and other non-traditional forms of religious life as a result of media content.

**Theorem 3.** *The revival of religiosity in the third decade is associated with new religions and alternatives to traditional religious activities.*

We demonstrated in Theorem 2 that religious revival occurs outside of the great churches. The table below shows that, following the initial increase in 2001, there is a decrease that continues to the present day. The increase in faith declarations appears only in religions and communities other than Christianity.

In terms of numbers, this means that the Catholic Church has lost over 300,000 worshippers in the last decade, while the A.C. Evangelical Church has lost 30,000 and the Greek Catholic Church has gained 12,000 worshippers. In comparison, 57,000 believers signed up for other confessions in 2021, a 33,000 increase in the number of people declaring their faith. For the first time, paganism and natural spirituality appear in statistics in 2021.

Proof of Theorem 3. In the context of the information given in Table 3, we can say that the revitalization of religiosity as one of the possible scenarios is reflected precisely in alternative religious movements but not in officially registered churches and communities. From a demographic perspective, the population in the Slovak Republic has kept stable since 2001 at 5.379 million. It was 5.398 million in 2011. Slovakia had 5.459 million people in 2020. We can say that the population of Slovakia is stable, with a slight upward trend. The decrease in the number of believers declaring their faith or church affiliation is thus not due to a decrease in the population. This is also supported by the Table 1 data, which show a significant increase in those claiming to be non-believers. We observe that the general trend in Slovakia is atheization and religious revitalization in religious communities that do not belong to the group of registered churches and religious societies. However, revitalization does not take place solely through the alternative activities of traditional churches (Cergetova Tomanova et al. 2021).

**Table 3.** Changes to the church's declaration of membership.

| Church/Year | 2001 | 2011 | 2021 |
|---|---|---|---|
| Catholic Church | +12.3% | −8.6% | −6.2% |
| Evangelical Church of the Augsburg Confession | +0.74 | −1.0% | -0.6% |
| Greek Catholic Church | +0.7% | −0.2% | +0.2% |
| Others | +0.1% | +0.3% | +1.2% |
| Believers in general | +11.2% | −8.1% | −6.2% |

**Theorem 4.** *The prospect of developing religiosity is in the pluralization of the religious environment.*

Based on previous findings, we assume that the revitalization of religiosity can take place in the pluralization of the religious environment and its opening up to new stimuli. This premise is based on the claims of several cited authors, who believe that new evangelization in traditional churches will lead to the restoration of spiritual life and religiosity (Hlad 2021). The new evangelization should make room for the younger generation's creativity and activities. If this were a road, we would see these activities in various associations and communities. Nothing like this occurs, as shown in the Table 4 below.

**Table 4.** Belonging to a religious association or community.

| Kind of Belief/Kind of Belonging | Do Not Belong | Belong Passive | Belong Active |
|---|---|---|---|
| Deeply Believe | 55.5% | 17.3% | 27.3% |
| Believe | 88.6% | 7.9% | 3.5% |
| Only for Tradition | 94.1% | 5.4% | 0.5% |

In general, the majority of respondents in our study did not belong to any religious organization or community—even more than half of those who declare that they are deeply religious. This means that their faith is restricted to religious acts related to the liturgy. Religion is a separate issue for them from their interests and daily lives. This situation can be attributed to the first decade of formalism, when it was socially acceptable to declare one's faith with this also being the only thing a man did in relation to religion (Pavlíková et al. 2021). On the other hand, formalism is also the result of a kind of vague guarantee, an insurance policy in the event of death, or, as we have already stated, in the case of services provided by churches. However, we do not have research data on this claim.

**Proof of Theorem 4.** The validity of the theorem on the need for religious pluralism has been partially confirmed. As we demonstrated in Theorem 3, the revival of religiosity has been confirmed only in unregistered communities, i.e., outside traditional and large churches. Revitalization in large churches is lacking because the number of believers is declining. We also do not see secularization because, as shown in Table 4, believers have little interest in participating in religious organizations (Kardis et al. 2021). Conversely, the decrease in the number of believers and increase in the number of non-believers indicate that atheization is the most significant trend that has not been mentioned in theoretical assumptions. □

## 3. Materials and Methods

The methodological starting point is a questionnaire study of Slovak religiosity conducted by the research agency Data Collect s. r. o. as part of the project APVV-17-0158. The basic ensemble consisted of the entire Slovak population. A selection pool of 1000 respondents of varying sex, age, residence, and education from all Slovak regions was drawn from the basic set using a quota system. Another point of departure were the entries obtained from the Slovak Statistical Office and the Czech Statistical Office. In the case of



the Slovak Republic's Statistical Office, we collected data from population, house, and apartment censuses in 1991, 2001, 2011, and 2021. In the case of the Czech Statistical Office, we used data from the 2021 house and apartment census. The obtained data were processed using correlation; statistical analysis focused on selected variables defined in theories and analyses of the decades studied. The procedures used have a quantitative feel to them. As a result, the results presented in the article are statistical in nature, and we were unable to address the causes of these changes in their interpretation, nor did we pay attention to the significant differences in what specific respondents represented by the terms "faith" and "confession", or what the content of their faith was (Stark and Glock 2012). Reconciling the terminology used by statistical authorities with that used in our research was methodologically challenging. There is a clear distinction between a person who does not believe and a man who refuses to declare his religion, or someone who believes but does not understand the subject of his religion. In this regard, additional research is required to unify the content of the terms used in order to achieve more concrete results.

## 4. Discussion

We can argue with presented scenarios, because what the cited text refers to as secularization is appropriate to refer to as atheization based on the findings of our investigation. The significant decline in Catholic Church membership over the last decade, as well as the significant increase in respondents identifying as non-believers, is evidence of atheism.

A religious stabilization or even revitalization model: the dynamic activity of religious organizations in various social environments is primarily responsible for the revitalization of religiosity (Mariański 2018). Furthermore, over half a million believers who do not belong to any registered church or community have been added in Slovakia in the last ten years. We can contrast the described situation with that of the Czech Republic, with which Slovakia was united until 1992. Another significant fact is that, over the last three decades, the number of people in the Czech Republic who refuse to answer the question about their religiosity has increased significantly. Up to 44.7 percent of the Czech Republic's population refuses to answer the religious question in the regular census.

We can assume that this is because they regard their religion as a personal, almost intimate thing that they do not want to declare or discuss in public. This is referred to as the privatization of religion. Religion is a private, rather than a public matter. In the Slovak Republic, however, the most significant increase is seen among those who do not practice any religion and are thus classified as atheists. We tested possible scenarios listed in the available literature based on available interpretations that focused on Slovakia and the prospects of religiosity. It turns out that there is not much truth to the scenario that the authors refer to as a secularization scenario. According to our findings, what Štefaňak or Mariańsky refer to as the secularization process is actually an atheizing process. Further investigation will be required to distinguish between two significant groups of respondents: a group of people who declare that they are non-religious, and those who refuse to express or communicate their religiosity. More research will be needed to determine whether respondents who claim to be non-believers do not belong to a religious group or association.

These are not forms of private religion, in the sense that they merely share the faith, whether expressed or not. Respondents who are hesitant to comment on a religious issue will also require special consideration. It will also be necessary to ascertain the reasons for their refusal to discuss their own religiosity (Tkacova et al. 2019). Although it appears that the most realistic prediction of the development of religiosity in Slovakia is a gradual increase in atheism, this assumption needs to be confirmed and supplemented with new data on the attitudes of non-believers and those who refuse to communicate about their religious beliefs.

## 5. Conclusions

The paths of religious development in Slovakia are impossible to predict. However, based on the results of our research, we can assume what the potential scenarios are for

the development of religiosity. These scenarios partly correspond to the assumptions made by sociologists dedicated to studying religiosity in Central Europe. Looking at the potential development of religiosity, we can talk about two separate lines. One is the line of development of Christianity as a historically significant religion that shaped the character of the state. The second line is religiosity as such, and tendencies among those residents who do not declare belonging to a registered church or refer to themselves as non-believers.

If we look at the possibilities available to Christianity through registered churches, we assume that the path of secularization will be the most likely. In Christian churches, not only is the number of believers declining, but so is the number of clerics who provide liturgy and other activities related to the active practice of the faith. More and more, the laity are taking on various roles in the church.

The decline in the number of believers in churches can be a sign of inner purification. Believers do not stay in the church because of tradition or out of fear of social exclusion. Young believers find their own ways of realizing their own religiosity, which is dominated by elements of contemporary popular culture, including the use of new media. Young believers are active in organizing various events and creating a new generation that is able to exist even in a pluralistic environment. Young Christians are aware of that. Unity does not lie in the fact that we are all the same, but in the fact that we all care about the common good. This path is a positive vision for Christian churches, because it is a vision of dynamic internal development. The future of Christian churches probably lies not in an increase in the number of believers, but in developing an active experience of faith. It is a revitalization, but not in the way that the cited authors, Mariaski and Štefaňák, envision it. Thus, it is revitalization in the sense of increasing the quality of living the faith among church members, rather than an increase in the number of believers. It is not about revitalization in the sense of a resurgence in the number of believers, but about revitalization in the sense of reinvigorating the faith.

If we follow the line of believers who do not subscribe to the church or to Christianity, we do not see an increase in belonging to any particular faith or creed. We cannot say that Christianity is being replaced by another religion. Rather, we could argue that those who rejected Christianity for various reasons are looking for various alternatives in the form of other religions and movements that are not tied to the historical heritage of the country. In our research, we did not follow the reasons why people abandon Christianity and look for alternative religions. We believe that new possibilities of communication play a large role, allowing people eager to meet their spiritual needs to look for an alternative. This path of religious development has been referred to as the pluralization of the religious environment. However, it cannot be seen as an alternative to Christianity.

**Author Contributions:** Conceptualization, P.K.; methodology, A.L.; validation, E.D.; formal analysis, E.D.; investigation, P.K.; resources, P.K.; writing—original draft preparation, P.K.; writing—review and editing, E.D.; visualization, P.K.; supervision, P.K.; project administration, A.L.; funding acquisition, A.L. All authors have read and agreed to the published version of the manuscript.

**Funding:** This research was funded by Cultural and Educational Grant Agency Ministry of Education, Science, Research and Sport of the Slovak Republic KEGA, grant number 032UKF-4/2021.

**Institutional Review Board Statement:** Not applicable.

**Informed Consent Statement:** Not applicable.

**Data Availability Statement:** Data is available here: https://www.ukm.ff.ukf.sk/wp-content/uploads/2022/09/TABS-VIERA-2020-10.xlsx.

**Conflicts of Interest:** The authors declare no conflict of interest. The funders had no role in the design of the study; in the collection, analyses, or interpretation of data; in the writing of the manuscript; or in the decision to publish the results.

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
