# Peer review of "Analysis of Religiosity in Slovakia since 1989 and Paths of Its Development"

_religions, doi:10.3390/rel14030415_

Round 1

Reviewer 1 Report

The paper represents a good and important theme of analyses of religiosity in Slovakia from post-communist perspective. The proposed title should be changed as the word "prediction" is not common for academic language. The suggestion would be to replace second part of the title with: the possible pathways and future developments or similar. The methodological part should be included at the beginning and not in the end of the paper. It should be clearer how author of the paper participated or did not participate in the mentioned research and its only using data. More extensive and narrative conclusion is needed. Numbering in the conclusion part does not read well and should be summary of main points and ways forward. If the paper is addressing future perspectives the conclusion needs to reflect that and not relay on someone else's work which is cited currently. 

Author Response

Dear reviewer, thank you for your feedback.

We changed prediction to pathways. We have changed the conclusions as per your instructions 496 - 647. The methodological part is moved to line 71. 

We provided the mentioned research, but because anonymity was preserved, we are unable to speak further..

Best regards, authors

Reviewer 2 Report

Overall I thought that this article add a coherent framework for studying religiosity (or the lack thereof) in Slovakia from 1989 to present. It’s fundamental argument that religious revival characterized its first postcommunist decade, followed by general weakening of religious identity during the second decade and a gradual stabilization during the third. I have no doubts about the statistics that the author has used to help define this periodization nor about the author’s mastery of the literature.

The author’s use of English is relatively adequate requiring relatively minor edits.

Nonetheless, there are aspects of this article that could use improvement. For instance, on the second page the author states that “Slovakia had a clergy fascist state during World War Two, the so-called first Slovak Republic. That is, a state in which the president was a priest and the politician was subservient to fascist Germany.” Why not just mention Josef Tiso directly? Without doing so, the outside reader cannot make much connection here between for example Slovakia and Croatia which had a similar fascist leadership under Ante Pavelić and Cardinal Stepinac his main clerical supporter. there was a revival of Stepinac and to some extent Pavelic in Croatia in the 1990s. Was it the same in Slovakia or not? if different how so? I know that discussions of fascism or fascist revivalism is probably a taboo topic in Slovakia, but it is interesting and helps draw in non-Slovak readers. Similarly the author mentions on the same page that “after 1989, Christian political parties were formed to create the impression that Christianity was returned to society as a morally restorative element that brings moral principles in order to society… They were known as the keepers of tradition and traditional ideals many of these ‘keepers of these ancient and traditional values,’ it was later revealed, aim to retain Russia's and its ideologies influence in Central Europe.” What political parties are you referring to and who exactly were these keepers of traditional values? And what specific connections do they have with Russian political ideologies? Again the readers have no idea where you're going here so you need to specify.

These criticisms fortunately require only revisions on the authors discussion of religious identity during Slovakia's first post-communist decade.

Author Response

Dear reviewer

thank you for your comments. We accept your comments and edited the text according to them. Lines 108, 113 - 117. We cannot upload the current version here, therefore, we attach it to the message to the editor. 

Reviewer 3 Report

As stated by the abstract, the article “presents the results of research into the current state of selected religiosity phenomena in Slovak republic and models possible pathways of future development of selected phenomena” (4). Though, the goal of the authors’ inquiry seems to be more delimited: “To see if the aforementioned processes of religious revitalization [shown by Štefaňak 2021’s study] can be recognized in Slovak society and if any sort of syncretism or other means of realizing one's own faith is part of this revitalization” (65).

As per the authors, a finding of their investigation is “the significant decline in Catholic Church membership over the last decade, as well as the significant increase in respondents identifying as infidels,” a fact interpreted as “evidence of atheism” (433) or, at least, a sign “of the development of religiosity” in this sense (463). Based on this statement, I would like to do some observations: a) To reduce the decline only to the Catholic church seems to attribute to one denomination a phenomenon that, as said in the article, is also seen in the Evangelical church (319, 322) and even in other undefined “religious societies” (397). In respect of what has been said during the presentation, it would be convenient to avoid this reductionism. b) Even if terminology can be a methodological challenge (482), it does not seem to be up-to-date to refer to those who may be called just unbelievers or “non-believers” (256, 268, 283) as “infidels” (91, 271, 395, 427, 456, 464). c) Moreover, it seems to me a further reductionistic to interpret as atheism various realities such as secularization, agnosticism, or religion fluidity.

If I may say so, the article would be much clearer if it were limited to presenting the results of the authors’ investigation, their interpretation, and the pertinent conclusions. During the results’ presentation, the intercalation of theorems based on researches conducted by other investigators that must be validated to finally reach a conclusion does not contribute to a fluent reading of the writing.

Even if an essay has explored different points of view, a good conclusion clearly shows the authors’ position. As I see it, this is not the case. Marianski and Štefaňak’s theories (489) are pointless in this part. Maybe, this paragraph should precede the previous references to the models (57 or 435) and be substituted by a proper conclusion.

Author Response

Dear reviewer, thank you for your comments on the article.

We used the term infidel instead of infidels: lines 121, 301, 433, 466, 473, 495, 502, 579.

We have developed new conclusions based on your recommendations. 572 - 610.

We send the edited article through the editor's account.

With thanks

authors

Round 2

Reviewer 3 Report

The authors have assumed the comments made in the Report 1 and have also improved their article introducing some appropriate corrections.